# Stable chaos and delayed onset of statisticality in unimolecular dissociation reactions

Sourav Karmakar[1], Pankaj Kumar Yadav[1] & Srihari Keshavamurthy [1]*

Statistical models provide a powerful and useful class of approximations for calculating reaction rates by bypassing the need for detailed, and often difficult, dynamical considerations. Such approaches invariably invoke specific assumptions about the extent of intramolecular vibrational energy flow in the system. However, the nature of the transition to the statistical regime as a function of the molecular parameters is far from being completely understood. Here, we use tools from nonlinear dynamics to study the transition to statisticality in a model unimolecular reaction by explicitly visualizing the high dimensional classical phase space. We identify generic features in the phase space involving the intersection of two or more independent anharmonic resonances and show that the presence of correlated, but chaotic, intramolecular dynamics near such junctions leads to nonstatisticality. Interestingly, akin to the stability of asteroids in the Solar System, molecules can stay protected from dissociation at the junctions for several picoseconds due to the phenomenon of stable chaos.

[1] Department of Chemistry, Indian Institute of Technology, Kanpur, Uttar Pradesh 208 016, India. *email: srihari@iitk.ac.in

A crucial requirement for the success and applicability of statistical rate theories is that intramolecular vibrational energy redistribution (IVR) occurs unhindered and sufficiently fast. In particular, the IVR timescale needs to be short compared to the typical vibrational timescale associated with the transition from activated complex to products. Such an assumption, along with the more global one of ergodicity or thermalization, implies that there are no preferred IVR pathways. However, several studies (see refs. [1–6] for recent reviews) indicate that, irrespective of the size and complexity of the system, all IVR pathways are not equivalent and deviations from the Rice-Ramsperger-Kassel-Marcus (RRKM) theory[7] predictions can occur even for sizeable molecules. Recent examples, highlighting the need for detailed insights on the dynamics of energy flow, include dynamical effects in nucleophilic substitution reactions[8,9], lack of thermalization in reactive intermediates[10,11], vibrational energy sequestration in activated bimolecular reactions[12], low barrier conformational reactions[13], reactions involving large amplitude motions[14], and mode-specificity in gas-surface reactions[15,16]. Thus, the dream of "molecular surgery", whether directly controlling the IVR[17] or circumventing it using ultrashort pulses[18,19], may not be that pessimistic after all. However, realizing the dream requires[19,20] identifying the specific vibrational modes that are involved in the dominant energy flow pathways. This, interestingly, relates back to an old and still unresolved question: what are the necessary and sufficient conditions for the validity of the RRKM model? The two issues are related since identifying the vibrational modes that efficiently couple to the reaction coordinate is an exercise in dynamics, and it is the same dynamics that ultimately validates the twin assumptions of RRKM—a sufficiently fast IVR, and non-recrossing of the transition state.

Addressing the above question requires models for IVR that consider the various anharmonic vibrational resonances at different levels of detail. The classical kinetic models[21,22] associate nonstatisticality with a partitioning of the phase space into dynamically distinct regions, implicitly due to specific resonances. On the other hand, the quantum state space based local random matrix theory (LRMT)[23,24] explicitly takes into account the anharmonic resonances and predicts a quantum ergodicity threshold[1,25], delineating the facile and restricted IVR regimes in a molecule. Given the classical mechanical underpinnings of RRKM theory, one expects that the roots of nonstatisticality are in the classical phase space and therefore connecting the classical and quantum models would yield a better understanding of the transition to statistical regime. At the same time, such a study would also highlight purely quantum effects that need to be considered for designing rational control fields. However, despite valuable insights being obtained in systems with two vibrational degrees of freedom ($f = 2$), progress has been slow for $f \geq 3$ due to the inherent challenges, technical as well as conceptual, posed by the increased dimensionality of the phase space[26–30]. A case in point are the pioneering computational studies[31,32] by Bunker on the dissociation dynamics of model triatomic molecules ($f = 3$)—one of the earliest attempts to correlate the validity of the statistical approximation with molecular parameters such as masses, dissociation energies, and vibrational frequencies. It took more than a decade before Oxtoby and Rice elegantly rationalised[33] Bunker's results using ideas based on nonlinear dynamical systems theory. However, the analysis utilized reduced dimensional $f = 2$ subsystems since analysing even the $f = 3$ case, to quote Oxtoby and Rice, "rapidly becomes very complicated even for quite small molecules". Extending the Oxtoby-Rice analysis to $f \geq 3$ is important, since one can then predict the onset of statisticality without performing extensive dynamical calculations. However, this has remained an outstanding challenge.

Here we take a first step towards such a goal by studying a $f = 3$ model inspired by Bunker. We go beyond the Oxtoby-Rice paradigm by constructing the network of nonlinear resonances, also known as the Arnold web, in different dynamical regimes. Our results highlight specific features in the phase space, called as resonance junctions, that bring out the crucial role of the third degree of freedom in the transition to statisticality. Although hints about the role of the junctions to the IVR process have been around for nearly three decades[34–44], up until now there has been no effort to ascertain their importance in reaction dynamics. Here we precisely achieve this by explicitly correlating the dynamics near the resonance junctions with unimolecular dissociation lifetime distributions. We argue that slowing down of chaos near the junctions leads to the delayed dissociation of the molecules and, ultimately, nonstatistical dynamics.

## Results

**Model Hamiltonian.** The Hamiltonian of interest

$$H(\mathbf{q}, \mathbf{p}) = \sum_{i=1}^{3} \left[ \frac{1}{2} G_{ii}^{(0)} p_i^2 + V_i(q_i) \right] + \epsilon \sum_{i<j=1}^{3} G_{ij}^{(0)} p_i p_j \quad (1)$$

is identical to the one used by Oxtoby and Rice[33], where the coordinate dependent $G$-matrix elements in the original Bunker model are replaced by their equilibrium values $G_{ij}^{(0)}$. The form of the Hamiltonian above involves bond coordinates and arises naturally in the local mode representation. Such Hamiltonians are known to be ideal for investigating the dynamics of anharmonic oscillators. Note that the local mode representation is equivalent[45] to the anharmonic normal mode representation, wherein the momentum coupling terms transform to potential coupling terms. The assumption of equilibrium $G$-matrix allows us to use analytic forms for action-angle variables to gain insights into IVR dynamics en route to the dissociation. The factor $\epsilon \in (0, 1)$ in front of the coupling terms, not present in the earlier studies, allows us to systematically study the onset of statisticality. For $\epsilon = 0$ the modes are uncoupled and there is no IVR, whereas for $\epsilon = 1$ we recover the original system with the possibility of extensive IVR.

Following Bunker[32], the potential energies for the stretching modes ($i = 1, 2$) are chosen to be Morse oscillators $V_i(q_i) = D_i[1 - \exp(-\alpha_i(q_i - q_i^0))]^2$ and the bending mode ($i = 3$) is modelled by a harmonic oscillator, $V_3(q_3) = \omega_3^2(q_3 - q_3^0)^2/2G_{33}^{(0)}$. In what follows, we highlight the central results by choosing the parameters in Eq. (1) to correspond to Bunker's model number 6, which loosely represents the ozone molecule[32]. More specifically, for the model of interest the stretching modes have a dissociation energy of $D_1 = D_2 \equiv D = 24$ kcal mol$^{-1}$, the harmonic mode frequencies are taken to be $(\omega_1, \omega_2, \omega_3) = (1112, 1040, 632)$ cm$^{-1}$, and we focus on the dissociation dynamics at $E = 34$ kcal mol$^{-1}$ (See Supplementary Table 1 for the full list of parameter values).

**Survival probability and lifetime distributions.** Initial conditions $(\mathbf{p}_0, \mathbf{q}_0)$ satisfying $H(\mathbf{p}_0, \mathbf{q}_0) = E = 34$ kcal mol$^{-1}$ were propagated up to a final time of $T = 40$ ps with the condition $q_1(t)$ or $q_2(t) > 7.5$ au signalling a dissociation event (See Supplementary Methods and Supplementary Table 2 for details). We compute the lifetime distribution

$$P(t) = -\frac{1}{N(0)} \frac{dN(t)}{dt} \equiv -\frac{d}{dt} S(t) \quad (2)$$

and the survival probability $S(t)$ as a function of $\epsilon$ in order to identify the transition to the RRKM regime. In the above $N(t)$ is

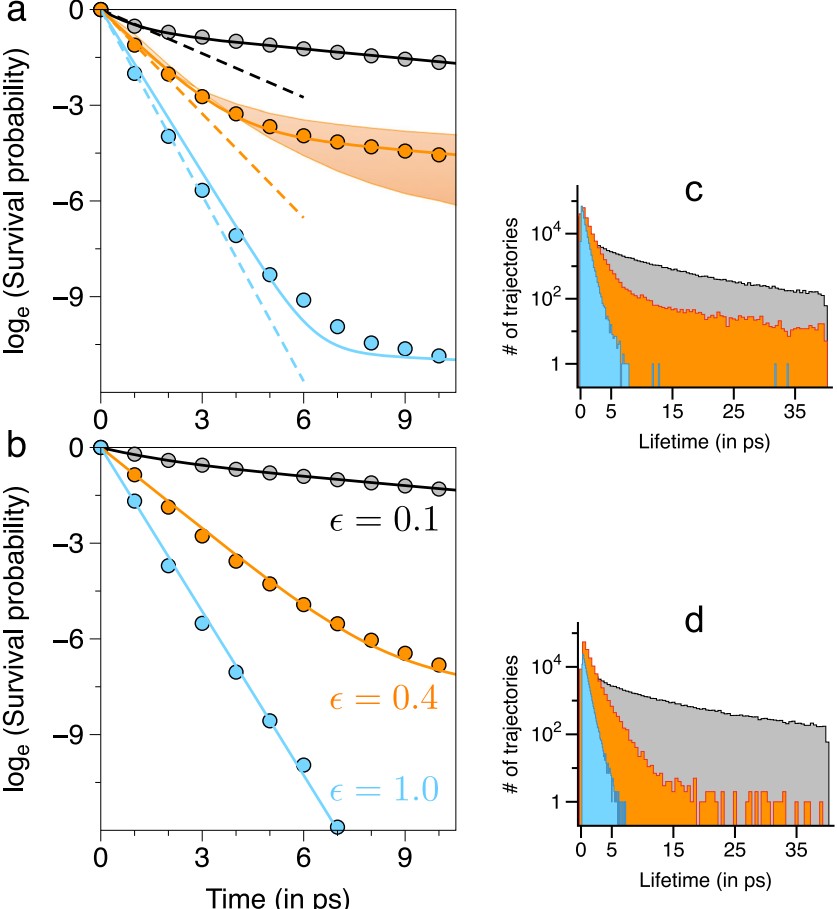

**Fig. 1 Survival probabilities and lifetime distributions at $E = 34$ kcal mol$^{-1}$. a** Computed survival probability (filled circles) as a function of $\epsilon$ for initial conditions satisfying $H(\mathbf{p}_0, \mathbf{q}_0) = E$. **b** As in **a** for initial conditions satisfying $H(\mathbf{J}_0, \boldsymbol{\theta}_0) = E$ chosen from the $(\theta_1, \theta_2, \theta_3) = (\pi/2, \pi/2, 0)$ slice of the phase space. The solid lines are multi-exponential fits to the data and the dashed lines (shown only in **a**) are the short time single exponential fits to the data. In **a** the variations with different choices of the phase space slices for $\epsilon = 0.4$ are shown as a shaded band. **c, d** Lifetime distributions on a linear-log scale for cases **a** and **b**, respectively.

the number of active molecules that remain undissociated at time $t$. Assuming the validity of RRKM, the survival probability exhibits exponential behaviour $S(t) = e^{-k(E)t}$ with $k(E)$ being the microcanonical rate constant. It is useful to point out that although the specific assumptions inherent to RRKM lead to the exponential law[46], observation of an exponential behaviour need not guarantee rates in accordance with the RRKM prediction[1]. Note that in our computations we discard trajectories from the initial ensemble that do not dissociate until the final time and convergence studies were done by varying the ensemble sizes (Supplementary Fig. 1).

In Fig. 1a we show the survival probability for select values of $\epsilon$ and, as expected[31], the transition to statisticality is nearly complete by $\epsilon = 1$. For smaller values of $\epsilon$ a multi-exponential behaviour can be seen, indicating two or more timescales, consistent with the observed long time tails for $P(t)$. Further insights can be obtained by choosing initial conditions on various angle slices of the energy shell $H(\mathbf{J}, \boldsymbol{\theta}) = E$ (Supplementary Methods and Supplementary Table 2). We anticipate that quantitative differences in $N(t)$ and $S(t)$ for different angle slices would signal nonergodicity and, as seen later, allow us to visualize the key phase space features that regulate the dissociation dynamics. In Fig. 1b the results for an example phase space slice $(\theta_1, \theta_2, \theta_3) = (\pi/2, \pi/2, 0)$ are shown. Comparing to Fig. 1a it is clear that for $\epsilon = 0.4$ and $1.0$ there are significant differences, with the latter case exhibiting a nearly complete transition to

statisticality, as also evident from the lifetime distribution. Thus, the significant variations in the $\epsilon = 0.4$ decay for different angle slices seen in Fig. 1a indicates nonstatistical dynamics. Note, however, that the source of the nonstatistical behaviour is unclear at the moment.

At this stage one can only guess that the nonstatistical behaviour for $\epsilon < 1$ is due to insufficient overlap of resonances[33]. This, in turn, in LRMT is related[25] to the local density of resonantly coupled states being below a certain threshold. Alternatively, the results in Fig. 1 are fit well by a bi-exponential form (Supplementary Methods, Supplementary Fig. 2 and Supplementary Tables 3, 4), suggesting[22] that the phase space is partitioned into two dynamically distinct regions. As we show below, neither of the above arguments can explain the nonstatistical behaviour for $\epsilon > 0.2$. In fact, as we establish next, an explicit knowledge of the resonances and their connectivity is required to ascertain the extent of resonance overlap and unambiguously identify the dominant resonances that potentially partition the phase space.

**Visualizing the transition to statisticality.** To address the above issues, we take a direct approach and map the Arnold web i.e. network of resonances in the high dimensional phase space. We use, among several available methods[47], the fast Lyapunov indicator (FLI) approach[48]. The FLI technique is ideal for identifying chaotic, resonant, and non-resonant dynamics using short time

trajectories (Supplementary Methods, Supplementary Figs. 3,4 and Supplementary Note 1).

In order to appreciate the expected structure of the Arnold web, we transform to action ($\mathbf{J}$) and angle ($\boldsymbol{\theta}$) coordinates and express the Hamiltonian (Supplementary Note 2) as $H(\mathbf{J}, \boldsymbol{\theta}) = H_0(\mathbf{J}) + \epsilon V(\mathbf{J}, \boldsymbol{\theta})$ with the zeroth-order part given by

$$H_0(\mathbf{J}) = \sum_{k=1,2} \omega_k J_k \left(1 - \frac{\omega_k}{4D_k} J_k\right) + \omega_3 J_3 \qquad (3)$$

where, $\omega_k \equiv \sqrt{2D_k \alpha_k^2 G_{kk}^{(0)}}$ are the harmonic frequencies of the stretches. The modes are coupled by the perturbation

$$\begin{aligned} V(\mathbf{J}, \boldsymbol{\theta}) = &\sum_{l,m=1}^{\infty} f_{lm}^{(12)}(\mathbf{J})[\cos(l\theta_1 - m\theta_2) - \cos(l\theta_1 + m\theta_2)] \\ &+ \sum_{l=1}^{\infty} g_l^{(13)}(\mathbf{J})[\sin(l\theta_1 - \theta_3) + \sin(l\theta_1 + \theta_3)] \\ &+ \sum_{m=1}^{\infty} g_m^{(23)}(\mathbf{J})[\sin(m\theta_2 - \theta_3) + \sin(m\theta_2 + \theta_3)] \end{aligned} \qquad (4)$$

with $f_{lm}(\mathbf{J})$ and $g_l(\mathbf{J})$ being functions of the parameters of the Hamiltonian in Eq. (1) (See supplementary Note 2 for the expressions for the Fourier coefficients). Note that for the model of interest the maximum value of the stretching actions $J_{max} \sim 15$, beyond which the modes dissociate.

The terms in Eq. (4) allow us to identify the resonances that are central to the IVR process. For example, the condition $l\dot{\theta}_1 \approx m\dot{\theta}_2$ with $(l, m)$ being coprime integers signals a $l : m$ resonance $l\Omega_1 \approx m\Omega_2$ between the nonlinear (anharmonic) frequencies $\boldsymbol{\Omega}$ of the two stretching modes. More generally, resonances $l\dot{\theta}_1 + m\dot{\theta}_2 + n\dot{\theta}_3 = 0$ with $l, m, n \in \mathbb{Z}$ (set of integers, positive, and negative) will be denoted as $(l, m, n)$ and said to be of order $\mathcal{O} = |l| + |m| + |n|$. While resonances $(l, m, 0), (l', 0, m')$ and $(0, l'', m'')$ indicate IVR involving any two of the modes, $(l, m, n)$ with non-zero entries implies active energy sharing between all the three modes. The resonance conditions are satisfied for certain values of actions, representing a surface in the action space. For $f = 2$ the resonance surfaces intersect the constant energy surface $H = E$ at isolated points. However, for $f \geq 3$ the resonance surfaces intersect $H = E$ to form an intricate connected network, owing to which the nature of phase space transport is fundamentally different with the number of possible IVR pathways being far greater than in systems with $f < 3$. A characteristic feature on the Arnold web for $f \geq 3$ is the existence of multiplicity-$r$ resonance junctions, with $r \leq (f - 1)$, formed by the intersection of $r$ independent resonances. A junction formed by the intersection of the two independent resonances $(l, m, n)$ and $(l', m', n')$ will be denoted by $\mathcal{M}_{l',m',n'}^{l,m,n}$. Note that an infinity of resonances emanate from a junction. For example, at the junction $\mathcal{M}_{0,l',-m'}^{l,0,-m}$ the condition $\mu(l\Omega_1 - m\Omega_3) + \nu(l'\Omega_2 - m'\Omega_3) \approx 0$ with integers $(\mu, \nu) \neq 0$ is also satisfied.

In Fig. 2a we show the computed Arnold webs as a function of the coupling strength $\epsilon$ for $E = 34$ kcal mol$^{-1}$. The webs are computed for initial conditions on the phase space slice $(\theta_1, \theta_2, \theta_3) = (\pi/2, \pi/2, 0)$, in order to compare with Fig. 1b, and are projected on the two dimensional $(J_1, J_2)$ space. We stress here that the web features are weakly dependent on the angle slice for $\epsilon \ll 1$ corresponding to near-integrable regimes. On the other hand, with increasing $\epsilon$ the system becomes nonintegrable and, expectedly, there is a strong angle dependence. Hence, as discussed in the next section, different slices can reveal additional structures. For $\epsilon = 10^{-4}$ one can observe the various nonlinear resonances as lines of varying widths. The

stretch-bend resonances $(l, 0, -m)$ and $(0, l, -m)$ show up as vertical (fixed $J_1$) and horizontal (fixed $J_2$) strips, respectively. The stretch-stretch resonances $(l, -m, 0)$ appear as lines with positive slopes. That the resonances are dense on the Arnold web can be seen in Fig. 2c. Most of the phase space exhibits quasi-regular dynamics and some of the high order resonances for large stretch excitations have overlapped leading to chaotic dynamics (indicated as yellow coloured regions in Fig. 2a). A prominent feature (indicated by green arrow in Fig. 2a) is the presence of the stretch-bend resonances $(1, 0, -1)$ and $(0, 1, -1)$ involving both the stretching modes. These two resonances intersect around $J_1 \approx J_2 \approx 6.3$, forming a multiplicity-2 junction $\mathcal{M}_{0,1,-1}^{1,0,-1}$ (indicated by a blue circle). Many such junctions $\mathcal{M}_{0,l',-m'}^{l,0,-m}$ exist and typically, as can be clearly seen in Fig. 2c, junctions involving different order resonances abound in the action space. As $\epsilon$ increases the resonances become wider and by $\epsilon = 10^{-2}$ overlap significantly leading to chaotic dynamics. Nevertheless, even for $\epsilon = 0.1$ the 1:1 stretch-bend resonances, although partially broken, persist. In addition, structures in the low (high) stretching (bending) excitation regions, shown in Fig. 2d and Fig. 2e, exhibit intertwined regular and chaotic dynamics. For coupling strengths beyond $\epsilon \sim 0.1$ the Arnold web structure is lost, with the system being in the Chirikov regime (See Supplementary Note 4), and replaced by a set of fragmented features embedded in a sea of chaos. These fragmented features exist over a finite range of angles and, therefore, different angle slices may reveal some residual structure.

In Fig. 2b we show the dissociation lifetimes for initial conditions on the slice in order to correlate the dissociation dynamics with the structures on the Arnold web. For $\epsilon = 0.1$ fairly long lifetimes are seen for initial conditions in the vicinity of the partially broken 1:1 stretch-bend resonances, near the junction $\mathcal{M}_{0,1,-1}^{1,0,-1}$ and, near regions of low (high) stretch (bend) excitations. In contrast, for $\epsilon = 0.4$ and 1.0, most of the trajectories dissociate within $\sim 2$ ps with some of the longer lifetime trajectories for $\epsilon = 0.4$ being concentrated near $\mathcal{M}_{0,1,-1}^{1,0,-1}$. Further confirmation comes from analysing the dynamics on the so called zero-momentum surfaces (Supplementary Note 3, Supplementary Fig. 8).

**Validity of the Oxtoby-Rice and the kinetic models.** Since, strictly speaking, there are no isolated resonances on the Arnold web, the application of the resonance overlap criterion[33] is not straightforward. Nevertheless, given the dominance of the $(1, 0, -1)$ resonance, one can estimate a threshold value of $\epsilon \sim 0.2$ for widespread chaos, agreeing with the results shown in Fig. 2a (Supplementary Note 4). Given that the analysis is independent of the angle slice, the clear multi-exponential decay seen in Fig. 1a even for $\epsilon \sim 0.4$ shows that the requirement of widespread chaos is not enough to ensure statistical behaviour.

From the perspective of the kinetic model[22], Fig. 2a suggests that the resonances $(1, 0, -1)$ and $(0, 1, -1)$ result in two dynamically distinct phase space partitions—an excited bending region and regions corresponding to excited stretches. The bi-exponential decays in Fig. 1 can then be ascribed to the slow IVR between the two regions. However, the partitioning resonances are broken around $\epsilon \approx 0.2$ and yet the survival probability continue to exhibit multi-exponential behaviour. We now show, as hinted by Marcus, Hase, and Swamy[22], that highly correlated intramolecular motion leading to infrequent transitions between qualitatively different types of dynamics occurs with significant consequences for the IVR and the subsequent dissociation dynamics.

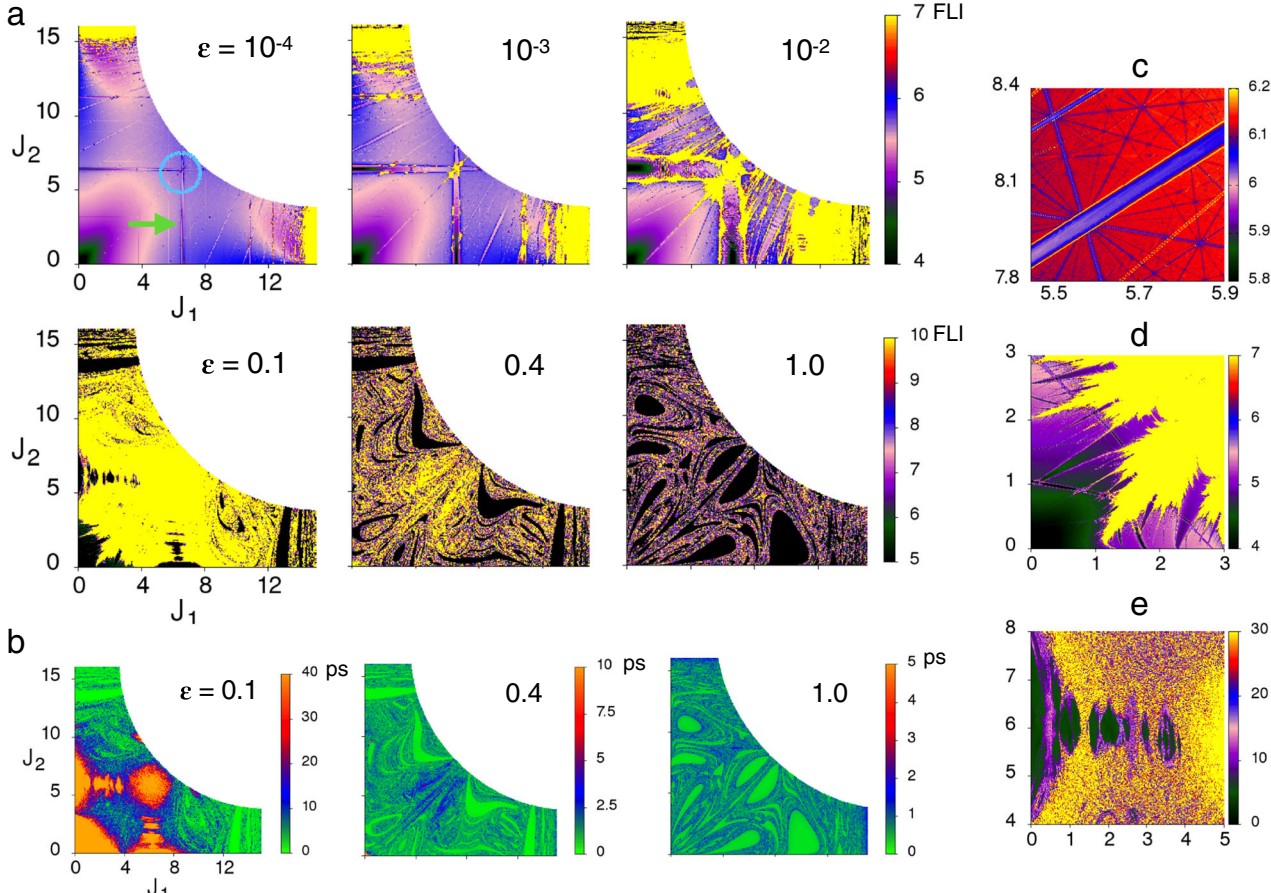

**Fig. 2 Evolution of the Arnold web as a function of the coupling strength at** $E = 34$ **kcal mol**$^{-1}$. **a** Arnold web, projected on to the $(J_1, J_2)$ action space, as a function of increasing coupling strength $\epsilon$ for initial conditions on the phase space slice $(\theta_1, \theta_2, \theta_3) = (\pi/2, \pi/2, 0)$ propagated upto a final time $T = 40$ ps. Yellow regions indicate chaos. For $\epsilon = 10^{-4}$ case, a prominent 1:1 stretch-bend resonance (green arrow) and a junction (blue circle) formed by the intersection of two such independent resonances are highlighted. **b** Trajectory lifetimes (in picoseconds) corresponding to the initial conditions on the Arnold web in **a** for varying $\epsilon$ values. **c** Zooming into a portion of the Arnold web for $\epsilon = 10^{-3}$. **d**, **e** As in **c** for $\epsilon = 0.1$. Note the different FLI scales. All the plots shown here are computed on a 500 × 500 uniform grid of initial conditions. Note that all FLI values greater than or equal to the maximum scale indicated are shown in yellow.

**Role of the resonance junctions**. Interestingly, Fig. 2b shows that for $\epsilon = 0.1$ long lifetimes trajectories exist around the $\mathcal{M}_{0,1,-1}^{1,0,-1}$ junction despite Fig. 2a exhibiting essentially chaotic dynamics around the same region. There are other junction regions in Fig. 2b displaying similar behaviour. However, here we focus on the prominent $\mathcal{M}_{0,1,-1}^{1,0,-1}$. We make two remarks in order to understand this observation. Firstly, trajectories initiated on a specific angle slice are not constrained to that slice. Secondly, the definition of the FLI implies that the trajectory is chaotic when integrated for sufficiently long times $T$. However, any trapping that may have occurred at intermediate times $0 \leq t \leq T$, on potentially a different angle slice, is not apparent from the final value of the FLI alone (See Supplementary Fig. 7b for an example). At the same time, given that initial conditions with similar FLI values exhibit significantly different lifetimes, the fact remains that the data in Fig. 2 establishes the existence of chaotic trajectories that get trapped around certain regions in the multi-dimensional phase space. Thus, it is essential to explore the FLI features on different slices to identify the source of the long lifetime regions seen in Fig. 2b. Toward this end, in Fig. 3a, b we show the webs for the original slice $(\pi/2, \pi/2, 0)$ as well as a different angle slice $(\pi/2, 0, 0)$ over an expanded FLI scale. Note that such structures are seen over a range of angle variables (Supplementary Figs. 9,10) and the two slices shown here are

representative of the key structures seen on the web. In addition, in Fig. 3c, d we show the zoomed FLI map near the junctions on both the slices. Note that although the concentric iso-FLI regions observed in Fig. 3c are not as prominent as in Fig. 3d, the FLIs do have particularly large values in the vicinity of the junction. A key point to note here is that despite the large FLI values, indicating chaotic motion, Fig. 3e, f shows that the trajectories in the vicinity of $\mathcal{M}_{0,1,-1}^{1,0,-1}$ have fairly long dissociation lifetimes. Based on the remarks regarding the FLI made above we can rationalise the observations by associating the longer lifetimes near junctions with partially chaotic trajectories that are under the influence of both $(1, 0, -1)$ and the $(0, 1, -1)$ resonances.

In order to confirm the arguments above we start by showing in Fig. 4a the lifetime distribution for initial conditions within specific concentric FLI-shells, as observed in Fig. 3d. Clearly, as we move away from the center of the junction the lifetime distribution peaks at shorter times, indicating the decreasing influence of the junction. Note, however, the long time tails in Fig. 4a and that even for initial actions sufficiently far away from the junction the distribution peaks around ∼ 5 ps. Nevertheless, the question remains as to whether such trapping near $\mathcal{M}_{0,1,-1}^{1,0,-1}$ is responsible for the second longer timescale in the survival probability in Fig. 1b. The answer to the question is in the affirmative. Firstly, the influence of the junction shown in Fig. 4a

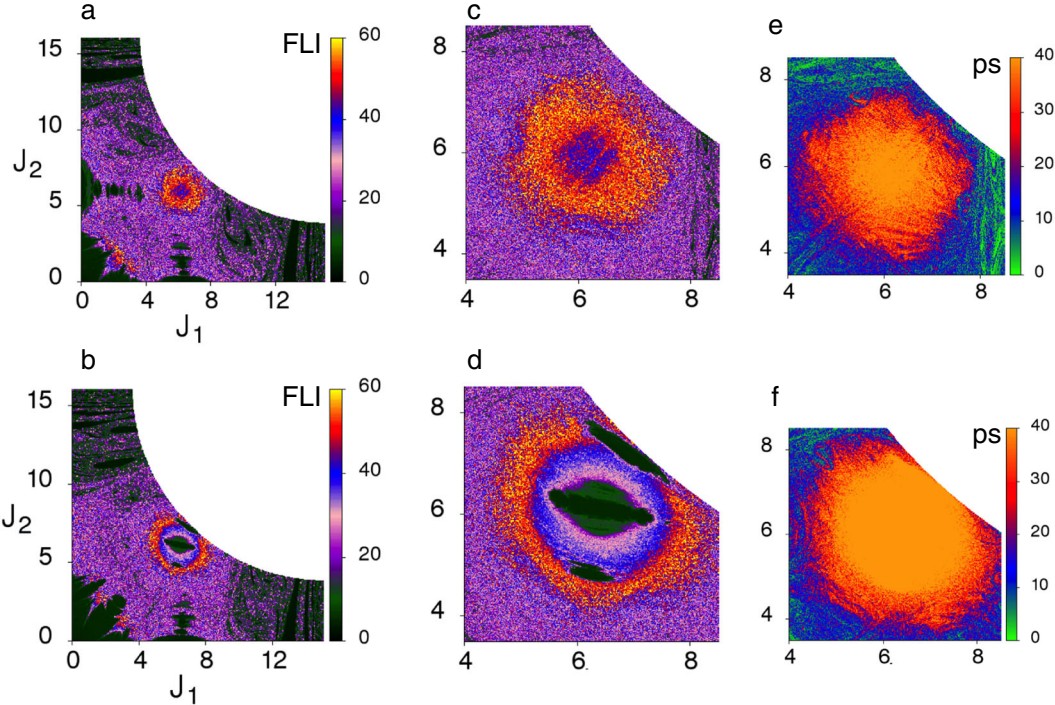

**Fig. 3 Details of the $\mathcal{M}_{0,1,-1}^{1,0,-1}$ junction at $E = 34$ kcal mol$^{-1}$. a, b** Arnold webs for $\epsilon = 0.1$ corresponding to the angle slices $(\theta_1, \theta_2, \theta_3) = (\pi/2, \pi/2, 0)$ and $(\pi/2, 0, 0)$, respectively, shown on an extended FLI scale. **c, d** Zooming into the junction for cases **a, b**, respectively. **e, f** Dissociation lifetimes in the vicinity of the junction for the two angle slices of interest.

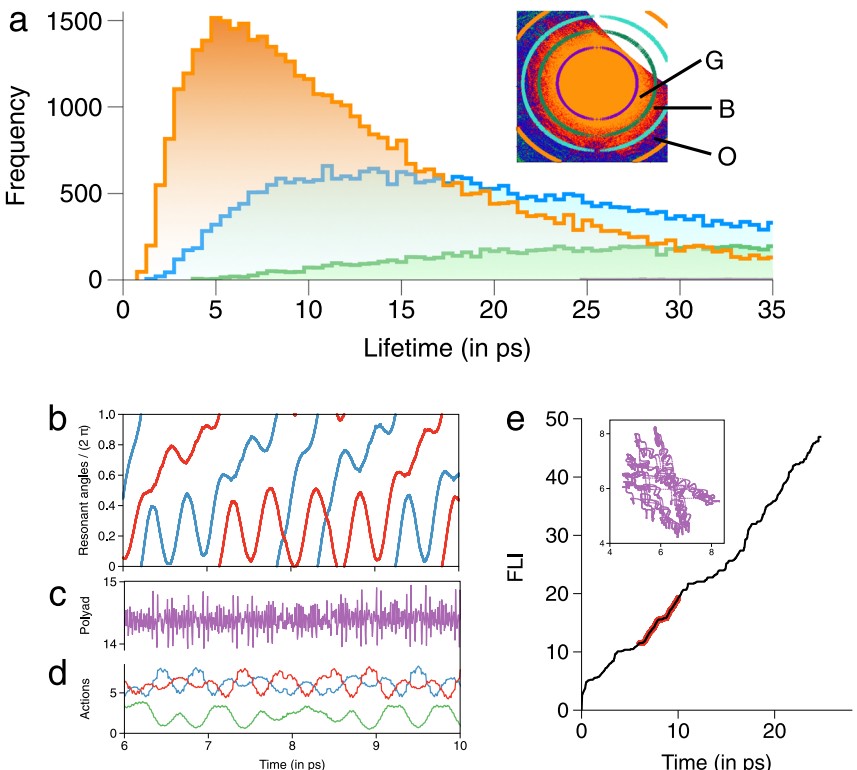

**Fig. 4 Influence of the $\mathcal{M}_{0,1,-1}^{1,0,-1}$ junction at $E = 34$ kcal mol$^{-1}$. a** Distribution of the lifetimes for initial conditions on the $(\pi/2, 0, 0)$ slice near the junction. Colours green, blue, and orange correspond to initial conditions chosen in the annular regions denoted G, B, and O as shown in the inset. Note that the initial conditions inside the region circumscribed by the purple circle remain undissociated until $T = 40$ ps. **b, c,** and **d** show the time evolution of the resonant angles $\theta_1 - \theta_3$ (blue) and $\theta_2 - \theta_3$ (red), polyad $(J_1 + J_2 + J_3)$ (purple), and the actions $J_1$ (blue), $J_2$ (red), $J_3$ (green) respectively for a trajectory initiated near the junction on the $(\pi/2, \pi/2, 0)$ slice of the web. **e** FLI versus time for the example trajectory. The portion highlighted in red corresponds to the timescale shown in **b, c,** and **d**. Inset shows the trapping around the junction in $(J_1, J_2)$ space.

is qualitatively similar for other choices of the angle slice as well. Secondly, our analysis reveals that initial conditions starting in the vicinity of the junction in Fig. 3c exhibit chaotic, but trapped, dynamics. This, given the definition of the FLI, it not entirely counterintuitive. As an example, in Fig. 4d we show the time evolution of the zeroth-order actions during a typical trapping episode for a trajectory starting from the vicinity of the junction on the $(\pi/2, \pi/2, 0)$ slice. That the trajectory is chaotic is evident from Fig. 4e showing the time evolution of the FLI. Nevertheless, over the entire time interval the actions are oscillating about a bounded region, undergoing highly correlated intramolecular dynamics. The trapping near the junction can also be seen in the $(J_1, J_2)$ space projection shown in the inset to Fig. 4e. Additional support comes from the near constancy of the quantity $(J_1 + J_2 + J_3)$ seen in Fig. 4c and the slow variation of the resonant angles $(\theta_1 - \theta_3)$ and $(\theta_2 - \theta_3)$ shown in Fig. 4b. Clearly, the results shown in Fig. 3 and Fig. 4, with similar influence due to the other junctions (Supplementary Figs. 11, 12), indicate that the trapped chaotic trajectories near junctions lead to the multi-exponential survival probabilities observed in Fig. 1b. Below, using time–frequency analysis, we show that the arguments survive angle-averaging and hence strengthen the case for linking the trapping near junctions to the observed nonstatisticality in Fig. 1.

The above, apparently puzzling, behaviour has been observed in studies on many different dynamical systems. Similar observations have been made in detailed studies of IVR in highly excited OCS molecule[38,40]. In dynamical astronomy the phenomena is known as stable chaos[49] and involves partially chaotic orbits near junctions[50], with key role played by the three body resonances[51]. Alternative explanations involving the sticking of chaotic trajectories[52] near partially regular structures have also been given, with connections to the concept of vague tori[53] that have been invoked in earlier studies[54] on unimolecular dissociation dynamics. However, a recent work[55] makes it clear that the mechanism of stickiness is expected to be very different in $f \geq 3$ systems.

**Molecular significance**. The central message, therefore, of Fig. 4 is that the trajectories trapped near junctions on molecularly significant timescales lead to the long time tails in the lifetime distributions (See Supplementary Note 5 and Supplementary Figs. 11, 12 for additional examples). At a junction the intramolecular dynamics is highly correlated with all three modes of the molecule sharing energy without any hindrance. However, the energy sharing happens on a fast enough timescale ($\sim 0.2$ ps near $\mathcal{M}_{0,1,-1}^{1,0,-1}$, as also seen from Fig. 4d) that the stretching modes get stabilised without dissociating for several picoseconds, and ultimately leading to nonstatisticality. Given the correspondence $J_k \leftrightarrow (n_k + 1/2)\hbar$ between the zeroth-order classical actions and the vibrational quantum numbers, a point in the action space (quantum number space) represents a potential initial zeroth-order bright state that might be accessed by experiments. A possible signature of being trapped near a junction would then be associated with the existence of an approximate polyad. For instance, trapping near $\mathcal{M}_{0,1,-1}^{1,0,-1}$ involves three-mode resonances and would imply approximate conservation of the $n_1 + n_2 + n_3$ polyad (Fig. 4c and additional examples in Supplementary Fig. 12). An example of such trapping may have been observed by Holme and Levine in their detailed computational study of IVR in the acetylene molecule[56].

**Persistence of the effect of junctions beyond the resonance overlap regime**. The role of the junctions for larger $\epsilon$ values can be further confirmed using the technique of wavelet-based[57] joint

time–frequency analysis (Supplementary Methods and Supplementary Fig. 5) to obtain the mode frequencies $\Omega_k(t)$ as a function of time. Apart from revealing the modes that are actively sharing energy in a specific time interval, such an analysis provides insights into the global phase space structures as opposed to the slices shown in Fig. 2a. We follow the IVR dynamics in the frequency ratio space (FRS)[36], $(f_1, f_2) \equiv (\Omega_1/\Omega_3, \Omega_2/\Omega_3)$ and construct a density map[42] by dividing the dynamically allowed range of the FRS into cells and recording the total number of visitations by trajectories in each cell up to a given time.

In Fig. 5a we show the FRS for trajectories that dissociate between $t \in (3.5, 4.5)$ ps, partly motivated from Fig. 1a, which shows the $\epsilon = 0.4$ case starting to deviate from a single exponential behaviour in this time interval. For $\epsilon = 0.1$ one can see enhanced density near several resonances and junctions, except near $\mathcal{M}_{0,1,-1}^{1,0,-1}$ and $\mathcal{M}_{0,2,-3}^{2,0,-3}$. This is consistent with the results in Fig. 4a since lifetimes significantly larger than $\sim 4.5$ ps arise due to the $\mathcal{M}_{0,1,-1}^{1,0,-1}$ junction. Nevertheless, it is clear that the Arnold web structure has not yet entirely crumbled for $\epsilon = 0.1$. For $\epsilon = 0.4$, Fig. 5a shows that very little of the web structure remains, but a clear enhancement near $\mathcal{M}_{0,1,-1}^{1,0,-1}$ can be observed, in agreement with Fig. 2b which shows lifetimes in the selected time range near the same junction. Although not clear from the figure, there is also enhanced density near the $\mathcal{M}_{0,1,-1}^{2,0,-3}$ and $\mathcal{M}_{0,2,-3}^{1,0,-1}$ junctions. In contrast, despite poor statistics, for $\epsilon = 1$ only $\mathcal{M}_{0,1,-1}^{1,0,-1}$ seems to influence the dynamics. Representative trajectories exhibiting trapping near junctions are shown in terms of the time evolution of the frequency ratios (Fig. 6a), in the FRS (Fig. 6b) and the corresponding FLI as a function of time (Fig. 6c). One can observe trapping times of about 2 ps ($\sim$40 bending vibrational period) with the FLI clearly levelling off near the junctions, signalling a "slowing down of chaos".

In contrast, the FRS for undissociated trajectories in Fig. 5b shows the increasing influence of the resonance junctions. The enhanced density regions for $\epsilon = 0.1$ occurs near several junctions, particularly near $\mathcal{M}_{0,1,-1}^{1,0,-1}$ and $\mathcal{M}_{0,2,-3}^{2,0,-3}$, which were absent in the case of Fig. 5a. For $\epsilon = 0.4$ we observe fewer junctions as compared to the $\epsilon = 0.1$ case. However, the resonances $f_1 = 1 = f_2$ and the junction formed by them still have a dominant effect on the dynamics. The region around the junction $\mathcal{M}_{0,2,-3}^{2,0,-3}$, also seen for $\epsilon = 0.1$, corresponds to the low stretching/high bending excitations. Interestingly, Fig. 5b shows that for $\epsilon = 1$ the only dominant influence is due to the $\mathcal{M}_{0,1,-1}^{1,0,-1}$ junction. Again, trajectories shown in Fig. 6d, e, f are representative of the dynamics of trajectories with long lifetimes. In particular, the trajectory for $\epsilon = 0.1$ is an instance of correlated intramolecular motion punctuated by infrequent transitions between dynamically distinct regions[22]. Our computations confirm that a significant number of such trajectories persist until $\epsilon \sim 0.4$ and are symptomatic of long dissociation lifetimes.

Despite the heterogeneity of the FRS due to the existence of distinct dynamical regions in the phase space, one cannot directly infer whether the enhanced density is due to frequent visitations or extended sojourn times of the trajectories. Thus, we quantify the extent of trapping near junctions by computing the distributions of longest locking and the total locking times (Supplementary Note 6). The distributions pertain to locking near any of the chosen (Supplementary Table 5) junctions and we do not attempt to dissect this further in terms of specific junctions. The results in Fig. 5c, d, e, f establish that there is significant trapping near the junctions. Moreover, the shift of the maximum to $\sim 3$ ps for the total locking time distribution in Fig. 5d indicates several locking events experienced by the trajectories.

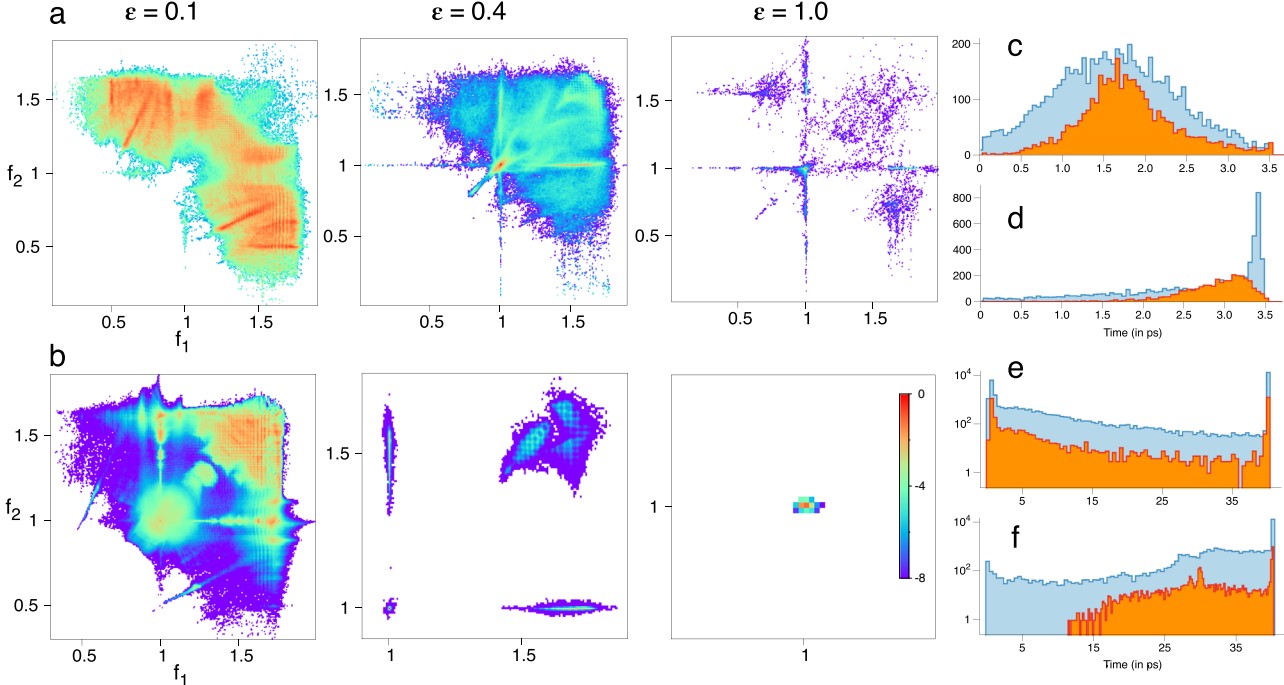

**Fig. 5 Time–frequency analysis results at $E = 34$ kcal mol$^{-1}$. a** Total number of visitations (normalized) in the nonlinear frequency ratio space $(f_1, f_2) \equiv (\Omega_1/\Omega_3, \Omega_2/\Omega_3)$ for trajectories dissociating between $(3.5, 4.5)$ ps for varying coupling strengths $\epsilon$. **b** As in **a**, but for trajectories that remain undissociated upto 40 ps. Note that the plots are on a $\log_e$ scale and locking near a junction $\mathcal{M}_{0,l',-m'}^{l,0,-m}$ implies $(f_1, f_2) = (m/l, m'/l')$. **c**, **d** Distributions of the longest locking times and the total locking times, respectively, near various junctions for $\epsilon = 0.1$ (blue) and $\epsilon = 0.4$ (orange) corresponding to trajectories dissociating between $(3.5, 4.5)$ ps. **e**, **f** Same as in **c**, **d** for trajectories that remain undissociated upto 40 ps. Note that in **e**, **f** the y-axis are on a log scale.

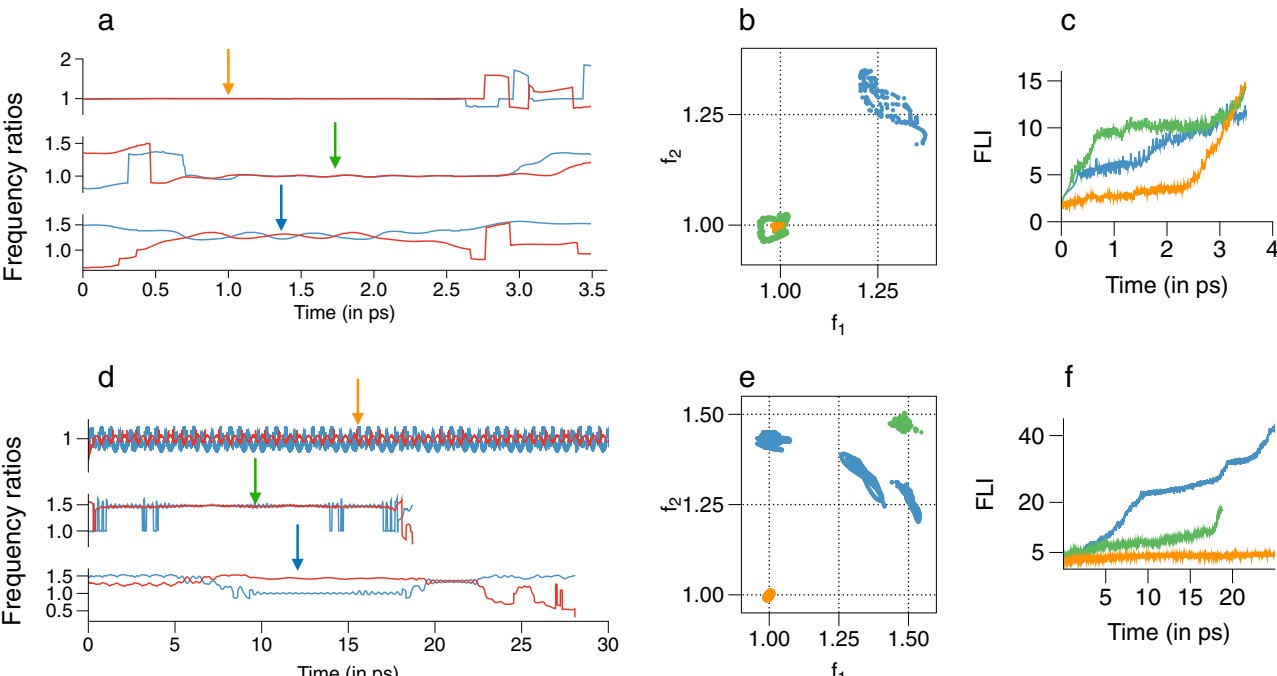

**Fig. 6 Representative examples for trajectories exhibiting trapping near junctions at $E = 34$ kcal mol$^{-1}$. a** Frequency ratios $f_1$ (blue) and $f_2$ (red) versus time of three sample trajectories that dissociate between $(3, 5, 4.5)$ ps. The trajectories correspond to $\epsilon = 0.1$ (blue arrow), $\epsilon = 0.4$ (green arrow), and $\epsilon = 1.0$ (orange arrow). **b** Data in **a** shown in the frequency ratio space during the trapping events. **c** Fast Lyapunov indicator (FLI) as a function of time for the three sample trajectories. **d**, **e**, and **f** show the corresponding plots for long lifetime trajectories.

## Discussion

We have established that the deviations from statisticality in gas phase unimolecular dissociation reactions are associated with dynamical stabilisation that occur near resonance junctions. Although the role of the junctions has been highlighted here for a specific Bunker model at a particular energy, our preliminary studies show that the junctions play an important role at different energies (Supplementary Fig. 13) as well as in other Bunker models, including model 6 without the assumption of equilibrium $G$-matrix elements (See Supplementary Note 7 and Supplementary Figs. 14, 15). Thus, we expect that stabilisation near resonance junctions should be a key factor in understanding the origins of nonstatistical reaction dynamics in more general systems as well. The junctions, which can only manifest in high dimensional phase spaces, are like "waypoints" in the energy flow traffic, wherein IVR is highly correlated and facile. Perhaps an analogy is worth mentioning at this juncture. The dynamical stabilisation in Fig. 4 due to extensive energy delocalisation is analogous to the textbook example of significant stabilisation of carbocations due to extensive charge delocalisation. In the former case the delocalisation occurs due to the presence of multiple resonances near a junction and in the latter case it has to do with the existence of several equivalent resonance structures.

Previous studies have stressed the need to analyse the nature of the Arnold web[39,58], role of the junctions[39,59], and local instability of the dynamics[60–62] to understand the transition to the RRKM regime. The present work brings these various viewpoints together in terms of identifying the resonance junctions as the critical feature, which potentially play the role of "hubs" that slow down global IVR. Note that the stabilisation near junctions and the consequent dynamical correlations are absent in simple tier models. However, the effects may be implicitly present in the LRMT formulation in terms of coupling chains involving off-resonant states[25,63]. In systems with $f > 3$ we expect that the hubs will lead to dynamical decoupling of a subset of vibrational modes from the rest on timescales of molecular significance. Indeed, the resonance junctions might justify an earlier speculation[60] by Kosloff and Rice regarding "interceptor processes" which result in a system behaving as if the energy is localised. The recent study[64] on the unimolecular dissociation of the dioxetane molecule appears to be a promising candidate in this regard.

We expect[42,44] that the dynamical stabilisation will survive quantisation. However, the extent to which quantum effects like dynamical tunnelling[65,66] can lead to enhanced localisation or de-trapping from the junctions[44,67] requires a systematic study of the classical and quantum dynamics near the junctions, particularly those with multiplicities greater than two. Such studies, given the modest effective dimensionality of the vibrational state space even for large molecules[68], may prove important towards the possibility of control by nudging the system to the regions of stable chaos using weak external fields[69–71].

## Methods

**Model parameters and details of the classical trajectory calculations**. See Supplementary Methods, Supplementary Fig. 1, and Supplementary Tables 1–2.

**Multi-exponential fits to the survival probabilities**. See Supplementary Methods, Supplementary Fig. 2, and Supplementary Tables 3–4.

**Computation and characterisation of the Arnold webs**. See Supplementary Methods, Supplementary Figs. 3–4, 7–10, 13–15, and Supplementary Notes 1–4,7.

**Wavelet time-frequency analysis**. See Supplementary Methods, Supplementary Figs. 5–6, 11–12, Supplementary Table 5, and Supplementary Notes 5–6.

## Data availability

The data that support the findings of this study are available within the article and its Supplementary Information files. All other relevant source data are available from the corresponding authors upon reasonable request.

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

## Acknowledgements

This work was supported by the Science and Engineering Research Board of India (EMR/2016/0062456). S. Ka. thanks the University Grants Commission (UGC), India for a doctoral fellowship. We acknowledge the IIT Kanpur High Performance Computing center for providing computing time.

## Author contributions

S. Ke. designed the research and wrote the paper. Computations and writing of the supplementary section were done by S. Ka. and P.K.Y. Analysis of the data was done by S.Ke., P.K.Y., and S. Ka.

## Competing Interests

The authors declare no competing interests.
