## [Peer Review File · Communications Chemistry]

Reviewers' comments:

Reviewer #1 (Remarks to the Author):

In this manuscript, the authors investigated a model by Oxtoby and Rice whose parameters are chosen from one of the Bunker's model. The model contains a parameter ϵ which controls the strength of coupling between the three modes. The authors showed that, if the parameter value of ϵ is less than 1, at a certain energy, the survival probability becomes non-exponential and also depends on the initial condition distribution. To understand the behavior, the authors investigated phase space structures of the model by computing Arnold webs, trajectory lifetimes, and time frequency map. Through the computation, the authors illustrated the non-exponential, initial-condition dependence of survival probability is due to the stable chaos occurred in neighborhoods of resonance junctions. Their findings are novel and invaluable but may be specific to the model and thus I think that this paper should be published in more specialized journal like the Journal of Chemical Physics.

Followings are specific comments to the authors:

1. In abstract, in the line 11, the authors claim "We identify generic features" but as far as I understand they simply investigate one specific model of certain parameter values and at a certain energy. Just working on the specific model, it is not clear if features that the authors identified are generic or not. In addition, the model Hamiltonian (1) does not seem natural because the g -matrix is set to be constant, the modes couple only through kinetic coupling, which are not usually the case.
2. In this manuscript, the authors investigate the angle slices $(\pi/2, \pi/2, 0)$ and $(\pi/2, 0, 0)$. As illustrated by the authors, just like the case between Fig.2a and Fig.3a (Arnold webs, $\epsilon = 0.1$), the structure of Arnold web depends on the slice. Since the author only show the two slices, it is not clear if Arnold webs shown in Fig.2a are generic features of the model at an energy.
3. In the line 206, if one of l, m, n is zero, this does not imply active energy sharing between all three modes.
4. In the line 235, The authors wrote "The stretch-bend resonances $(l, 0, -m)$ and $(0, l, -m)$ show up as vertical (fixed J_1) and horizontal (fixed J_2) strips, respectively.", but I guess that should be "The stretch-bend resonances $(l, 0, -m)$ and $(0, l, -m)$ show up as horizontal (fixed J_2) and vertical (fixed J_1) strips, respectively."
5. In the line 259, FLI is only an indicator of Arnold web and the fact that FLI does not have any apparent structure does not imply there is no Arnold web.
6. In the line 264, "slices slice" -> "slices".
7. In the line 309, the authors wrote "In order to resolve this apparent contradiction...", Fig.2b is on the phase space slice $(\pi/2, \pi/2, 0)$ and thus the Arnold web on the same slice should explain the result. The Arnold web on the different slice is that of different phase space region and how can that explain Fig.2b ?
8. In the line 343, did the authors estimate actual time scale for the energy sharing ? The authors might want to provide an estimate for that.
9. In the line 347, this naive correspondence may only hold for integrable systems.
10. Overall, the authors mainly focus on the energy transfer mechanism but to understand its relation to the reaction, its relation to the cylindrical manifold (reactive island) is important

because, if the trajectory is outside of the cylinder, it never reacts no matter how high the reactive mode energy is. The authors might want to discuss their results from the perspective of the cylindrical manifold.

Reviewer #2 (Remarks to the Author):

The manuscript reports results of a computational study examining the classical dynamics of a three degree of freedom coupled oscillator system that can undergo dissociation. Coupling among the oscillators is varied with a tunable parameter, and features of dynamics, in particular near resonance junctions, are examined in detail. The authors identify the influence of resonance junctions on trajectories that lead to dissociation, for which they find long dissociation lifetimes, an example of "stickiness", but, as they point out, of a different origin than the stickiness that has been well studied in smaller systems. They make contact between the detailed analysis they present on their three-degree of freedom system and studies of vibrational dynamics of the OCS molecule, as well as results of dynamical astronomy. As noted, they clarify important differences in the origin of stickiness near resonance junctions of systems with at least three degrees of freedom, and well studied two-degree of freedom systems, where stickiness of chaotic trajectories near regular islands in phase space has been well characterized. As a result, they make the case that in even highly excited polyatomic molecules, there are dissociative trajectories that will generally survive for a number of vibrational periods, so that deviations from RRKM theory predictions can be generally expected. This is an important result. The manuscript will interest many readers of Communications Chemistry and publication is recommended after the authors address the following:

The authors sell short the impact of their study with the very first sentence of the manuscript. There they state the oft-repeated and misleading claim that deviations from RRKM theory predictions occur when the IVR rate is slower than the reaction rate. Technically this is true but, again, it is misleading. Such a statement offers a deceptively safe space for RRKM theory. A reaction time may be ns, or ms, or days, and of course if the IVR time takes longer than such a reaction time the predictions of RRKM theory will not match the observed kinetics. That, of course, almost never happens. Still, RRKM theory may fail in those cases, and often does. RRKM theory predictions fail if the IVR time is slower than the vibrational time scale for transition from activated complex to product. This point is discussed in detail in Ref. 2 (page 474), among other sources, and deviations of observed rates of many chemical reactions involving sizable molecules from RRKM theory predictions due to the discrepancy between IVR time and the time for passage from an activated complex to product, again, a vibrational time, are documented in Ref. 2 and elsewhere. Therefore, the long-lived reactive trajectories near resonance junctions that the authors identify in their study will likely lead to deviations from RRKM theory predictions. Such "stickiness" slows down IVR on the vibrational time scale, and will thus influence the reaction kinetics.

We would like to thank both the referees for their critical reading of our manuscript and the encouraging comments. Based on the referees' comments and suggestions we have slightly modified certain parts of the manuscript, and these are indicated in red in the revised manuscript. Below we address the concerns of the referees.

1 Addressing the comments of Referee #1

In this manuscript, the authors investigated a model by Oxtoby and Rice whose parameters are chosen from one of the Bunker's model. The model contains a parameter ϵ which controls the strength of coupling between the three modes. The authors showed that, if the parameter value of ϵ is less than 1, at a certain energy, the survival probability becomes non-exponential and also depends on the initial condition distribution. To understand the behavior, the authors investigated phase space structures of the model by computing Arnold webs, trajectory lifetimes, and time frequency map. Through the computation, the authors illustrated the non-exponential, initial-condition dependence of survival probability is due to the stable chaos occurred in neighborhoods of resonance junctions. Their findings are novel and invaluable but may be specific to the model and thus I think that this paper should be published in more specialized journal like the Journal of Chemical Physics.

1. *In abstract, in the line 11, the authors claim "We identify generic features" but as far as I understand they simply investigate one specific model of certain parameter values and at a certain energy. Just working on the specific model, it is not clear if features that the authors identified are generic or not. In addition, the model Hamiltonian (1) does not seem natural because the g -matrix is set to be constant, the modes couple only through kinetic coupling, which are not usually the case.*

The generic features being referred to in the abstract are the resonance junctions. There is no doubt that the junctions (of various multiplicities) are generic in nonlinear Hamiltonian systems with three or more degrees of freedom. [See for example, G. Haller, *Chaos near Resonances*, Chapter 4, Springer-Verlag (1999) or S. Wiggins, *Chaotic Transport in Dynamical Systems*, Chapter 6, Springer-Verlag (1992)]. Thus, in the reaction dynamics of highly excited molecules, where anharmonicities are key to the IVR process, resonance junctions are expected to be present and influence the dynamics. Presently, precious little is known about the classical dynamics near different types of junctions. Moreover, which among the multitude of junctions are going to play a crucial role in the corresponding quantum dynamics is, as of yet, not known.

However, we interpret the referee's concerns as to whether the influence of the junctions, and their association with non-RRKM behaviour, are specific only to the model, energy, and the Hamiltonian considered in our work. We believe that this is not the case. As already mentioned in the main text, as long as junctions are present they are expected to influence IVR. For example the work of Shojiguchi et al (Ref. 39) and that of Honjo and Kaneko (Ref. 59) highlight the role of junctions in very different model systems. Similarly, our earlier work (Ref. 42) on the classical and quantum IVR dynamics in the effective Hamiltonian model for SCCl_2 has shown the influence of resonance junctions. Note that even for the simple model Hamiltonian considered in our present work, an a priori prediction of the value of ϵ for the system being in RRKM regime is an unsolved problem.

At the same time we appreciate the referee's concerns. We therefore have performed additional preliminary calculations which indicate that our detailed observations on model 6 are more general. More specifically:

- We have presented results for $E = 34 \text{ kcal mol}^{-1}$ to bring out the connection between non-RRKM and trapping near junctions. We have now added a Supplementary Fig. 13 that shows the influence of the junction at several other energies.
- The use of the parameter ϵ with model 6 of Bunker is due to the fact that model 6 transitions to RRKM behaviour at $\epsilon = 1$. Thus, different ϵ values aid in understanding the reasons for non-RRKM behaviour.

There are other models studied by Bunker which are non-RRKM. In Supplementary Fig. 15 we now show two such models wherein the influence of the junction persists up to $\epsilon = 1$.

- The referee’s concerns about the model Hamiltonian being too simple is partially right. However, kinetic couplings arise rather naturally in local-mode Hamiltonians. Secondly, the simplified coupling structure allows us to utilize the action-angle variables to obtain detailed insights into the dynamics. The full Bunker model is not so easily amenable in this respect. Nevertheless, the OCS model that has been studied, and has the same Hamiltonian structure as the full Bunker model Hamiltonian, does show indications for the junctions to be important. These studies have been cited in the manuscript. Moreover, studies utilizing a variety of Hamiltonians (cited Refs. 34-44) that model many different molecules, do hint at the role of junctions to the dissociation dynamics. A detailed study of some of these models is currently being planned in our group. However, motivated by the referee’s concern, we show a preliminary result on the original Bunker model 6 (without assumption of equilibrium G -matrix elements) in Supplementary Fig. 14. The influence of junctions is clear in the zero-momentum surface representation. In addition, we have now included a Supplementary Note 7 to bring out the generality of our observations.

2. *In this manuscript, the authors investigate the angle slices $(\pi/2, \pi/2, 0)$ and $(\pi/2, 0, 0)$. As illustrated by the authors, just like the case between Fig.2a and Fig.3a (Arnold webs, $\epsilon = 0.1$), the structure of Arnold web depends on the slice. Since the author only show the two slices, it is not clear if Arnold webs shown in Fig.2a are generic features of the model at an energy.*

The referee’s concern is indeed genuine. However, in our investigations we have carefully looked at many angle slices for several parameter values and different models as well. In the case pertaining to Fig.2a and Fig.3a the two slices essentially represent the key structures seen at other values of the slicing as well. In the revised manuscript we have noted this and have now included Supplementary figures 9 and 10 which explicitly show the different angle slices.

3. *In the line 206, if one of l, m, n is zero, this does not imply active energy sharing between all three modes.*

Agreed. In the revised manuscript we have now made this more explicit.

4. *In the line 235, The authors wrote “The stretch-bend resonances $(l, 0, -m)$ and $(0, l, -m)$ show up as vertical (fixed J_1) and horizontal (fixed J_2) strips, respectively.”, but I guess that should be “The stretch-bend resonances $(l, 0, -m)$ and $(0, l, -m)$ show up as horizontal (fixed J_2) and vertical (fixed J_1) strips, respectively.”*

Actually, the statement as written is correct. Note that the resonance $(l, 0, -m)$ implies $l\Omega_1(\mathbf{J}) = m\Omega_3(\mathbf{J})$. Using the fact that $\Omega_k(\mathbf{J}) \equiv \partial H_0(\mathbf{J})/\partial J_k$, the resonance condition is obtained as

$$l\omega_1 \left(1 - \frac{\omega_1}{2D_1} J_1 \right) = m\omega_3 \quad (1)$$

The above yields the resonant action $J_1^r = \text{constant}$, and this corresponds to a vertical line in the (J_1, J_2) action space.

5. *In the line 259, FLI is only an indicator of Arnold web and the fact that FLI does not have any apparent structure does not imply there is no Arnold web.*

We agree and are aware of this. That is why in the main text we have augmented the FLI-based Arnold webs with the wavelet analysis. In that context, in the original manuscript (Line 441 in the revised manuscript) we already did mention that “The Arnold web structure has not yet entirely crumbled for $\epsilon = 0.1$.” In addition, the subtlety of interpreting FLI in such dissociating systems is emphasized in the Supplementary Note 1 as well. Note that the Chirikov analysis presented in Supplementary Note 4 indicates that large scale overlap

of resonances sets in beyond $\epsilon = 0.1$. As a result, most of the invariant Kolmogorov-Arnold-Moser (KAM) tori have disappeared and the action plane is covered mostly by chaos. It is in this sense that we refer to the Arnold web structure being lost **beyond** $\epsilon = 0.1$. Nevertheless, as this is an important point, and may not be immediately clear to the readers, we have now included additional comments in the revised manuscript to bring this out explicitly.

6. *In the line 264, "slices slice" \rightarrow "slices".*

Thank you. This has been corrected in the revised manuscript.

7. *In the line 309, the authors wrote "In order to resolve this apparent contradiction...", Fig.2b is on the phase space slice $(\pi/2, \pi/2, 0)$ and thus the Arnold web on the same slice should explain the result. The Arnold web on the different slice is that of different phase space region and how can that explain Fig.2b ?*

In general, initial conditions on a specific angle slice do not remain on the slice. Thus, trajectories will leave the slice on different timescales and explore other phase space regions i.e., different angle slices. For example, an initial condition in the slice corresponding to Fig.2a in the vicinity of $\mathcal{M}_{0,1,-1}^{1,0,-1}$ leaves the angle slice and can come close to a structure corresponding to the junction over a range of angles and get trapped temporarily. This then leads to longer dissociation lifetimes. The key point here is that the Arnold web is shown projected on the stretching action subspace. However, this subspace is not an invariant subspace. A more global representation comes from the wavelet analysis in terms of the frequency ratio space, which does not refer to a specific angle slice. The wavelet results presented in Fig.4 of the manuscript are angle averaged and clearly shows the influence of the junctions.

However, the referee's point is well taken, in that our explanation might not be sufficiently clear. In the revised manuscript we have now rewritten the arguments to bring out the essence - (a) FLI indicating chaos does not reveal the intermediate time trapping and (b) Identifying the trapping structures, given that trajectories leave the angle slice, requires exploring other angle slices.

8. *In the line 343, did the authors estimate actual time scale for the energy sharing ? The authors might want to provide an estimate for that.*

One can estimate the timescale of energy sharing around a junction by inspecting the example action variations with time shown in the Supplementary Fig.11. For example, around the $\mathcal{M}_{0,1,-1}^{1,0,-1}$ junction, typical energy flow timescales are about 0.2 ps. This corresponds to about 3 – 4 bending periods. However, the timescales can be slightly different around other junctions and at different energies. More detailed investigations of the IVR dynamics near different junction, involving various order resonances, is certainly important and this is currently the focus of ongoing studies in our group. We have, however, noted the typical timescale for the model and energy of interest (34 kcal mol⁻¹) in the revised manuscript.

9. *In the line 347, this naive correspondence may only hold for integrable systems.*

Indeed the stated classical action - vibrational quantum number correspondence strictly holds in the integrable limit. However, spectroscopists typically think of the prepared initial nonstationary state in terms of the zeroth-order basis [See for example the recent Perspective article by Herman and Perry, Phys. Chem. Chem. Phys. **15**, 9970 (2013)]. Thus, the excitation prepares states that carry sufficient oscillator strength and they are called as zeroth-order bright (or feature) states. Note that the terminology "overtone state" or "combination state" precisely refer to the nature of the initial state in terms of the zeroth-order vibrational quantum numbers. We have emphasized this in the revised manuscript.

10. *Overall, the authors mainly focus on the energy transfer mechanism but to understand its relation to the reaction, its relation to the cylindrical manifold (reactive island) is important because, if the trajectory is outside of the cylinder, it*

never reacts no matter how high the reactive mode energy is. The authors might want to discuss their results from the perspective of the cylindrical manifold.

We agree with the referee that cylindrical manifolds play an important role in the reaction dynamics. However, note that in this work we highlight the trajectories that do react (dissociate) and identify the resonance junctions as the crucial features that slow down IVR sufficiently to delay the onset of statisticality, as indicated by the survival probability and lifetime distributions. Therefore, all trajectories of interest are within the reactive cylinder. In fact, from the early work of De Leon, Berne, and others it is known that the reactive island structure even in a seemingly chaotic phase space can result in non-RRKM dynamics. The earlier work by Paškauskas, Chandre, and Uzer (J. Chem. Phys. **130**, 164105 (2009)) does identify manifolds associated with specific periodic orbits which mediate the dynamics. There is no doubt that such manifolds, related to the reactive cylinders, mediate the dissociation dynamics in our system as well. However, constructing the cylindrical manifolds (Normally Hyperbolic Invariant Manifolds or NHIMs, to be precise) and connecting them up to the nature of the Arnold web in the reactant well is a challenging, but important and outstanding, problem. The prerequisite for this is to characterize the transport on the Arnold web in different dynamical regimes inside the reactant well. See, for example the schematic Fig. 1 in Shojiguchi et. al. Commun. Nonlinear Sci. Numer. Simul. **13**, 857 (2008). This is our ultimate long-term goal.

2 Addressing the comments of Referee #2

The manuscript reports results of a computational study examining the classical dynamics of a three degree of freedom coupled oscillator system that can undergo dissociation. Coupling among the oscillators is varied with a tunable parameter, and features of dynamics, in particular near resonance junctions, are examined in detail. The authors identify the influence of resonance junctions on trajectories that lead to dissociation, for which they find long dissociation lifetimes, an example of "stickiness", but, as they point out, of a different origin than the stickiness that has been well studied in smaller systems. They make contact between the detailed analysis they present on their three-degree of freedom system and studies of vibrational dynamics of the OCS molecule, as well as results of dynamical astronomy. As noted, they clarify important differences in the origin of stickiness near resonance junctions of systems with at least three degrees of freedom, and well studied two-degree of freedom systems, where stickiness of chaotic trajectories near regular islands in phase space has been well characterized. As a result, they make the case that in even highly excited polyatomic molecules, there are dissociative trajectories that will generally survive for a number of vibrational periods, so that deviations from RRKM theory predictions can be generally expected. This is an important result. The manuscript will interest many readers of Communications Chemistry and publication is recommended after the authors address the following:

- 1. The authors sell short the impact of their study with the very first sentence of the manuscript. There they state the oft-repeated and misleading claim that deviations from RRKM theory predictions occur when the IVR rate is slower than the reaction rate. Technically this is true but, again, it is misleading. Such a statement offers a deceptively safe space for RRKM theory. A reaction time may be ns, or ms, or days, and of course if the IVR time takes longer than such a reaction time the predictions of RRKM theory will not match the observed kinetics. That, of course, almost never happens. Still, RRKM theory may fail in those cases, and often does. RRKM theory predictions fail if the IVR time is slower than the vibrational time scale for transition from activated complex to product. This point is discussed in detail in Ref. 2 (page 474), among other sources, and deviations of observed rates of many chemical reactions involving sizable molecules from RRKM theory predictions due to the discrepancy between IVR time and the time for passage from an activated complex to product, again, a vibrational time, are documented in Ref. 2 and elsewhere. Therefore, the long-lived reactive trajectories near resonance junctions that the authors identify in their study will likely lead to deviations from RRKM theory predictions. Such "stickiness" slows down IVR on the vibrational time scale, and will thus influence the reaction kinetics.*

We are thankful to the referee for pointing this out! Indeed, the key comparison here is between the IVR timescale that repopulates the transition state region and the timescale on which the activated molecules become products. The ergodicity assumption ensures that no initial condition on the constant energy surface is special. We have rewritten the introductory sentences to bring out this aspect clearly.

REVIEWERS' COMMENTS:

Reviewer #1 (Remarks to the Author):

The Authors addressed basically all of my concerns thoroughly although some challenges still remain. Now, I do recommend it to be published in this journal. The Authors might want to address the following point before publication.

* p.348 (in the main text),

Here, the Authors wrote "Firstly, trajectories initiated on a specific angle slice are not constrained to the that slice."

This is true and I do not expect trajectories initiated on a specific angle slice constrained to that slice. Each angle slice has codimension 3 in the 5-dimensional equi-energy space and chances that one trajectory starting from a specific slice hits to another different angle slice is almost zero no matter how long the time T is. My concern in 7 (in 1 Addressing the comments of Referee #1) is still not fully answered. I said "Fig.2b is on the phase space slice $(\pi/2, \pi/2, 0)$ and thus the Arnold web on the same slice should explain the result." because in that case at least it is sure that the same set of trajectories are considered.

Technical comments (in the main text):

* p.117, involve -> involves

* p.163, need -> needs

* p.351, the that -> that (or the)

We would like to thank referee #1 for persisting with his question regarding the influence of the Arnold web on the specific phase space slice to understand the multi-exponential survival probability decay. Based on this we have now added extra figures (Figs. 3a,c,e and Figs. 4b,c,d,e in the revised manuscript) to bring out the influence of the junction. In addition, an extra paragraph has been included, indicated in red in the revised manuscript. Below we address the concerns of the referee.

1 Addressing the comments of Referee #1

1. *The Authors addressed basically all of my concerns thoroughly although some challenges still remain. Now, I do recommend it to be published in this journal. The Authors might want to address the following point before publication.*

** p.348 (in the main text),*

Here, the Authors wrote "Firstly, trajectories initiated on a specific angle slice are not constrained to the that slice."

This is true and I do not expect trajectories initiated on a specific angle slice constrained to that slice. Each angle slice has codimension 3 in the 5-dimensional equi-energy space and chances that one trajectory starting from a specific slice hits to another different angle slice is almost zero no matter how long the time T is. My concern in 7 (in 1 Addressing the comments of Referee 1) is still not fully answered. I said "Fig.2b is on the phase space slice $(\pi/2, \pi/2, 0)$ and thus the Arnold web on the same slice should explain the result." because in that case at least it is sure that the same set of trajectories are considered.

We agree with the referee that the co-dimensionality of the angle space means that chances of a trajectory initiated from a specific slice to be in the vicinity of a different angle slice is small. In the revised manuscript we now show the zoom of the junction region on the angle slice $(\pi/2, \pi/2, 0)$ relevant to Fig. 1b and Fig. 2b. This indicates large FLI values and lesser structure than the different angle slice $(\pi/2, 0, 0)$. However, and this is the essence of the notion of stable chaos, the corresponding lifetimes are still fairly large. Basically, the trajectories are exhibiting bounded or partially chaotic motion. Large FLI means that a small variation of actions on the same angle slice will result in fairly different motion, but still trapped near the junction. The trapping feature is brought out on the $(\pi/2, \pi/2, 0)$ slice in Fig. 4b,c,d,e in the revised manuscript. This example trajectory, which is fairly typical of trajectories initiated near the junction, clearly brings out the trapping near the junction, resulting in long lifetimes and hence the observed multi-exponential decay of the survival probability observed in Fig. 1b.

2. *Technical comments (in the main text):*

** p.117, involve \rightarrow involves * p.163, need \rightarrow needs * p.351, the that \rightarrow that (or the)*

Thank you for pointing these out. We have corrected them except for the comment on line 163 - this, as per our understanding, requires no correction.

Reviewers' comments:

Reviewer #1 (Remarks to the Author):

In the comment, 1 Addressing the comments of Referee #1, the author wrote

"This example trajectory, which is fairly typical of trajectories initiated near the junction, clearly brings out the trapping near the junction, resulting in long lifetimes and hence the observed multi-exponential decay of the survival probability observed in Fig.1b."

This is an interesting observation of the authors but, to conclude the example trajectory is typical, it is better to be quantitative, like the FRS plots as the authors did in Fig.5. If the authors intended to use Fig.4b,c,d,e just indicate some possible mechanisms behind non-exponential behavior observed in Fig.1b, it may be okay but if that is not the case, more quantitative supports are favorable.

We would like to thank referee #1 for his further comments. The slight changes are indicated in red in the revised manuscript. Below we address the concerns of the referee.

1 Addressing the comments of Referee #1

1. *In the comment, 1 Addressing the comments of Referee 1, the author wrote*

“This example trajectory, which is fairly typical of trajectories initiated near the junction, clearly brings out the trapping near the junction, resulting in long lifetimes and hence the observed multi-exponential decay of the survival probability observed in Fig.1b.”

This is an interesting observation of the authors but, to conclude the example trajectory is typical, it is better to be quantitative, like the FRS plots as the authors did in Fig.5. If the authors intended to use Fig.4b,c,d,e just indicate some possible mechanisms behind non-exponential behavior observed in Fig.1b, it may be okay but if that is not the case, more quantitative supports are favorable.

We have shown the example trajectory to indicate that one of the dominant mechanisms behind the non-exponential comes from trapping near junctions. In particular, the results in Fig. 4 b,c,d,e are meant to illustrate the stable chaos aspect. The FRS shown later in Fig. 5 are angle-averaged and hence, given that trajectories starting on a specific slice are not constrained to that slice, provide the quantitative support. Clearly, there are aspects of the dissociation dynamics that warrant further detailed investigations. Such studies are currently underway in our group and will be reported in future publications.